# A Combined Near-Infrared and Mid-Infrared Spectroscopic Approach for the Detection and Quantification of Glycine in Human Serum

**DOI:** 10.3390/s22124528

**Published:** 2022-06-15

**Authors:** Thulya Chakkumpulakkal Puthan Veettil, Bayden R. Wood

**Affiliations:** 1Centre for Biospectroscopy, Monash University, Clayton, VIC 3800, Australia; thulya.chakkumpulakkalputhanveettil@monash.edu; 2Centre for Sustainable and Circular Technologies (CSCT), University of Bath, Bath BA2 7AY, UK

**Keywords:** serum proteomics, attenuated total reflection Fourier transform infrared (ATR-FTIR) spectroscopy, near-infrared spectroscopy, chemometrics, multimodal data fusion, glycine

## Abstract

Serum is an important candidate in proteomics analysis as it potentially carries key markers on health status and disease progression. However, several important diagnostic markers found in the circulatory proteome and the low-molecular-weight (LMW) peptidome have become analytically challenging due to the high dynamic concentration range of the constituent protein/peptide species in serum. Herein, we propose a novel approach to improve the limit of detection (LoD) of LMW amino acids by combining mid-IR (MIR) and near-IR spectroscopic data using glycine as a model LMW analyte. This is the first example of near-IR spectroscopy applied to elucidate the detection limit of LMW components in serum; moreover, it is the first study of its kind to combine mid-infrared (25–2.5 μm) and near-infrared (2500–800 nm) to detect an analyte in serum. First, we evaluated the prediction model performance individually with MIR (ATR-FTIR) and NIR spectroscopic methods using partial least squares regression (PLS-R) analysis. The LoD was found to be 0.26 mg/mL with ATR spectroscopy and 0.22 mg/mL with NIR spectroscopy. Secondly, we examined the ability of combined spectral regions to enhance the detection limit of serum-based LMW amino acids. Supervised extended wavelength PLS-R resulted in a root mean square error of prediction (RMSEP) value of 0.303 mg/mL and R^2^ value of 0.999 over a concentration range of 0–50 mg/mL for glycine spiked in whole serum. The LoD improved to 0.17 mg/mL from 0.26 mg/mL. Thus, the combination of NIR and mid-IR spectroscopy can improve the limit of detection for an LMW compound in a complex serum matrix.

## 1. Introduction

Serum is fundamental for blood and nutrient transport and is an important matrix to monitor the health status of an individual. Importantly it can contain a number of direct and indirect indicators of disease progression. In fact, serum proteomic profiling is a well-established tool to identify important biomarkers associated with various cancer types [1,2,3,4], diabetes [5], atherosclerosis [6], and neurodegenerative diseases [7]. A number of important diagnostic markers are found both in the circulatory proteome and the low-molecular-weight (LMW) peptidome. The detection of LMW compounds in the circulatory proteome is analytically challenging because of the high dynamic concentration range of constituent protein/peptide species in serum [8]. The blood serum proteome is a complex cluster of proteins with high molecular weight (HMW) fractions including albumin (66.5 kDa; 35–50 mg/mL), immunoglobulins such as IgG (160 kDa; 8–18 mg/mL), transferrin (76 kDa; 1.5–4 mg/mL), and lipoproteins and sparse low molecular weight fractions, such as cytokines, chemokines, and peptide hormones [8]. The dominant cell signaling proteins, cytokines and chemokines, constitute a class of compounds in small concentrations (<5 pg/mL) and are approximately 6 to 70 kDa [9] and 7 to 12 kDa [10] in molecular weight, respectively. This large dynamic range exceeds the analytical capabilities of traditional proteomics methods, making the detection of lower concentrations of serum proteins extremely challenging demanding extensive fractionation/depletion of the HMW fraction prior to analysis. To this end, ultrafiltration strategies have been employed for the removal of HMW fractions in serum. However, since albumin is a transport protein that binds a large variety of compounds including hormones, lipoproteins, and amino acids, depletion of albumin may result in the specific removal of low abundance cytokines, peptide hormones, and lipoproteins of interest [11].

The current gold standard for validating putative bio-markers are antibody-based assays, which are aiming for the development of highly sensitive and specific assays for quantifying proteins [12]. Nevertheless, technical and operational bottlenecks of this approach including the requirement of specific antibodies for proteins of interest, time-consuming sample preparation strategies, and measurement times are of major concern [13]. In particular, although multiplex protein immunoassays are able to measure multiple analytes and provide information regarding the heterogeneity of disease states [14], challenges such as appropriate assay format and configuration, generation and characterization of capture ligands, probable cross-reactivity between reagents, proper analytical validation, and operational and quality control of assay panels remains elusive [13,15].

Vibrational spectroscopy has proven to be an excellent and effective analytical tool for detecting and characterizing biological materials because it provides functional details on biochemical composition and molecular dynamics [16,17,18,19,20,21]. In particular, infrared spectroscopy has become an accepted tool for biomedical applications with many proof-of-principle studies showing high specificity and sensitivity for disease detection and classification [18,20,22,23,24,25,26,27]. IR transmission absorption spectroscopic analysis of serum in its native state could be hampered due to the enormous water content and important information concerning the amide I band of proteins can be obscured. This can be overcome by performing infrared analysis on dried serum sample deposits.

Specifically, ATR spectroscopy provides high sensitivity and specificity for the detection of protein parameters in blood and blood-derived components. The mid-infrared (MIR) spectrum of serum provides information about biochemical parameters and justifies the exploration of the simultaneous determination of additional parameters of clinical interest [28]. For instance, Roy et al. investigated the potential of ATR spectroscopy in discriminating hepatitis B and C infected serum samples from control groups and were able to identify unique marker bands associated with hepatitis infection [29]. Similarly, Butler et al. demonstrated a combinatorial approach between high-throughput ATR-IR spectroscopy and machine learning technology for rapid triage of brain cancer using serum samples [30]. This is considered the first prospective clinical validation study which is able to differentiate cancer and control patients at a sensitivity and specificity of 93.2% and 92.8%.

Near-infrared spectroscopy is a proven technique for identifying proteins in serum [31]. Unlike MIR spectra, which interrogate fundamental vibrational transitions, NIR spectra include the overtones and combination modes of the fundamental vibrational modes. However, because of the broad nature of the bands in NIR spectroscopy, specific band assignments are notoriously complex. This does not negate the ability of the technique to quantify analytes, especially when aided with robust chemometric tools that can find correlations in the broad bands. To date, no other proof-of-concept NIR studies have been applied to detect and quantify LMW biomarkers in serum. Hence, this study will be the first proof-of-concept study of its kind detecting LMW compound in serum using glycine as a model system. In this study, ATR-FTIR and NIR spectroscopy are combined to improve the detection limit of an LMW component in serum using glycine as a model compound. Previously, ATR [32], Raman [33], and ultra-fast 2D IR [34] spectroscopic analysis have been applied in combination with chemometric tools to explore LMW serum analytes. Ultra-fast 2D IR spectral acquisition is a relatively rapid technique; however, state-of-the-art instrumentation is expensive, large, requires constant expertise to maintain good quality spectra, and produces large, complex datasets [35]. As previously reported, the simultaneous use of data obtained by two different spectroscopic techniques can increase the classification accuracy [36,37]. The improvement in the LoD and RMSEP is the result of combining fundamental molecular transitions from MIR spectra and overtones and combination bands from the NIR spectra.

Since it is the simplest amino acid with a molecular weight of 75 Da, glycine was used as a model example for a low-molecular-weight compound. Previously ATR-FTIR spectroscopy has been applied to serum after that the removal of water, by drying, and separating out the HMWF, by centrifugal filtration, which led to an improvement in the sensitivity of the technique [32]. Herein, using dried serum samples we demonstrate that combining MIR and NIR spectral data improves the LoD. Using this approach, we were able to achieve an LoD of ~0.17 mg/mL in as-received serum. This is far less than the normal levels of glycine in a serum sample (~0.02 mg/mL) [38] and is a vast improvement on current detection technologies including antibody assays and liquid chromatography-tandem mass spectroscopy are able to detect within the physiological range (0.011–0.044 mg/mL) [34].

For the first time, we studied the potential of NIR spectroscopy in detecting and quantifying an LMW amino acid using a cost-effective, miniature hand-held NIR spectrometer. Moreover, by using a data fusion approach we determined whether the combinatorial approach improves the limit of detection (LoD) compared to using the individual modalities. The proposed strategy is relatively simple and rapid because it utilizes the same sample under the same conditions. Although ATR and NIR are known to be stand-alone techniques, the current proof-of-concept study shows that combining ATR and NIR spectral data obtained from the same sample improves the LoD of LMW amino acids in serum.

## 2. Materials and Methods

### 2.1. Specimen Preparation

The study was performed on serum from screened donors secured from the Australian Red Cross. Three serum biological replicates were analyzed in the study. Since it is the simplest amino acid with a molecular weight of 75 Da, glycine was used as the reference molecule and its analytical grade counterpart was purchased from Sigma-Aldrich (G7126, St. Louis, MO, USA). The whole serum samples were augmented with dynamic range of concentrations of glycine; 0, 0.01, 0.05, 0.1, 0.5, 1, 2.5, 5, 7.5, 10, 17.5, 25, and 50 mg/mL. Thirteen sample sets were generated from each individual biological replicate resulting in 39 sample sets. A glycine solution was prepared with a concentration of 200 mg/mL in 0.9% NaCl solution (Baxter, Deerfield, IL, USA). Similarly, individual samples of serum albumin and globulins (Sigma-Aldrich, USA) were prepared in 0.9% NaCl solution (Baxter, USA). Furthermore, 100 µL of serum samples containing varying concentrations of glycine was deposited on different coverslips (24 × 55 mm with a thickness of 0.12–0.17 mm made from high-quality glass were purchased from Matsunami, Bellingham, WA, USA) for spectroscopic analysis. Briefly, the samples were dried at room temperature for 30 min and further blow-dried with a hair dryer to remove the majority of bound water. The same sample sets were used to build both ATR and NIR spectral data sets.

### 2.2. Spectroscopic Analysis: NIR Measurement Using Miniature NIR Spectrometer

The spectral acquisition was performed using a miniature, portable and hand-held spectrometer (manufactured by Neospectra Si-Ware systems (Menlo Park, CA, USA)) commercialized by Hydrix MedTech (Melbourne, Australia) that incorporates an integrated spectral sensor in the NIR regime between 2500 nm and 1350 nm. The spectrometer has been previously applied to detect and quantify malaria parasitemia in dried [39] and aqueous blood [40]. Technically, the Michelson interferometer was fabricated monolithically by exploiting the Micro-Electro-Mechanical System (MEMS) technology, which enabled the micropatterning or micromachining of the prime functional part of the spectrometer. Briefly, the spectrometer is designed for diffused reflectance measurements with three tungsten halogen lamps and reflectors surrounding the bulbs to optimize power coupling. The average scan time for analyzing the sample was 2–3 s and the integration time was 14.1 ms with a resolution of 16 nm. The wavelength accuracy and repeatability were determined to be ±1.5 and ±0.15 nm for λ = 1400 nm up to an absorbance level of 0.5 AU, respectively. Each scan produced 141 points. The highest diffuse reflectance material (>99%), Spectralon, was used as the background for spectral acquisition. Spectral data were acquired using the spectrometer’s software (spectroMOST microTM, version 1.0) from Neospectra, Si-ware system. As stated above, 100 µL of serum samples containing varying concentrations of glycine was deposited on different coverslip substrates, and samples were kept for drying at room temperature for 30 min, and an air dryer was used to completely dry the samples.

### 2.3. MIR Spectroscopic Analysis Using ATR—FTIR Spectrometer

An Alpha II platinum ATR spectrometer (Bruker Corporation, Billerica, MA, USA) was used for the spectral measurements in the mid-infrared region (4000-800 cm^−1^). It features a monolithic diamond crystal, which is tightly soldered into a tungsten carbide platform. The spectral parameters that were set included a resolution of 8 cm^−1^, a background and sample scan of 128, and a zero-filling factor of 2. Spectral data were acquired using the spectrometer’s software (OPUS version 7.5) from Bruker Corporation, USA.

### 2.4. Data Analysis: Multimodelling of ATR-FTIR and NIR Spectral Data

Spectral data analysis was performed in MATLAB 2018a (MathWorks, Inc., Natick, MA, USA) with the PLS toolbox (Eigen vector research, Manson, OH, USA), a multivariate and machine learning tool, which provides a unified graphical interface. Initially, ATR and NIR spectroscopic data of glycine-spiked serum samples were analyzed in the individual regression models generated for each technique. Both PLS-R and principal component regression (PCR) were used in data fusion regression modeling. Prior to regression analysis, Savitzy-Golay derivatization (2nd derivative with polynomial order of 2 and smoothening window of 15), a weighted normalization (standard normal variate (SNV), mean-centering was applied to the data set. ATR spectra were acquired in OPUS format, whilst the NIR data were saved in SPECTRUM form. Later, the data were exported as CSV files for aiding the combined spectral analysis. Figure 1 depicts the conceptual representation of the experiment and associated chemometrics. For the regression models, a calibration data set and validation model (520 spectra, 26 samples together) were built along with an independent test set as the prediction set (260 spectra, 13 samples). The Kennard–Stone method is a well-established method for selecting a subset of samples, which uniformly covers the data set and includes exterior samples as the calibration set. The method is based on a Euclidian distance calculation, which means that the samples are selected in such a way that they will uniformly cover the complete sample space, reducing the need for extrapolation of the remaining samples [41]. The predictive quality of the generated model can be assessed by a statistical parameter, root mean square error of prediction (RMSEP). Herein, RMSEP is derived from an independent prediction set of samples that have known Y values. The independent validation data set was randomized, loaded separately into the prediction test set, and totally independent of the training set.

## 3. Results

### 3.1. Multimodal Analysis of Glycine Deposits

Figure 2a represents the combined spectral analysis of dried glycine deposits (200 mg/mL) recorded on the stainless-steel substrate measured using NIR (Figure 2b) and ATR (Figure 2c) spectroscopy. For the normalized, second derivative ATR spectra, the important spectral information can be observed at 906, 1036, 1111, 1330, and 1414 cm^−1^. These specific features of glycine are attributed to the presence of CH_2_ rocking, C-N stretching, NH_3_ rocking, CH_2_ wagging, and O-C=O symmetric stretching in the spectra, respectively [32]. The bands at 1330 and 906 cm^−1^ are relatively intense along with several other bands due to aggregation and/or crystallization effects upon drying. The significant spectral characteristics are visible in the 1500–800 cm^−1^ region. Similarly, the NIR region exhibits stronger and more well-defined peaks at 2110, 2204, and 2375 nm. The peak at 2110 nm corresponds to an (N-H + C-H) combination band, and the peaks at 2204 nm and 2375 nm are assigned to combinations of (C-H and C-H) stretching modes, which are shifted from the literature values [42]. Similar bands can be attributed to the presence of proteins as well.

### 3.2. Spectral Analysis of Spiked Serum Samples: NIR and MIR Spectroscopy

Screened donor serum samples spiked with varying concentrations of glycine were analyzed using NIR (Figure 3) and ATR spectroscopy (Appendix A). Figure 3 shows the NIR spectra of glycine spiked into serum with a concentration series ranging from 0–50 mg/mL, whilst Figure 3b shows a magnified view of the 2500–2000 nm region. In Figure 3b, it is evident that the spectral intensity increases with spiked glycine concentration in the 2500–2000 nm region. The respective bands from glycine moieties are visible in the 2500–2000 nm region. Moreover, this region is known for the combinations of the stretching and deformation modes of protein [43]. The band at 2055 nm corresponds to the symmetric N-H stretch/amide Ib combination band [42]. Similarly, the band at 2180 nm is tentatively assigned to the N-H bend second overtone, C-H stretch/C=O stretch combination, and the C=O stretch/amide IIIb combination mode [42]. These bands are rather difficult to observe in the raw spectrum of glycine-spiked human serum samples and hence a second derivatization was applied to detect the presence of LMW compounds. Moreover, the dynamic concentration range of glycine dominates the relevant region (2500–2000 nm), which may obscure serum protein bands in the baseline-corrected raw spectra. Comparably, the band positions in the mid-infrared region are depicted in Appendix A. The glycine bands at 1330, 906, and 884 cm^−1^ are correlated with the incremental increase in concentration. Major band positions observed from the NIR and MIR spectra are represented in Appendix A.

### 3.3. Regression Analysis

Both PLS-R [21] and PCR approaches are well-known multimodal spectral analytical tools that are applied to extract information from complex data sets [44]. Generally, both methods aim to reduce dimensionality or solve multicollinearity problems. PCR projects the x-data onto a lower-dimensional space, either based on maximizing variance described in the x-data, whilst PLS-R is based on maximizing the covariance between the *x*-data and *y*-data. A literature survey indicates that the PLS-R outperforms PCR or at least provides better performance in detecting and quantifying molecules as PCR often requires extracting more components to achieve the maximum predictive ability compared to PLS-R [45]. However, herein we are investigating the prediction capabilities of unsupervised PCR and supervised PLS-R approaches. Different parameters or regions have been considered to build robust regression models. Initially, the robustness of individual MIR or NIR spectroscopic techniques was analyzed using PCR and PLS-R. Appendix A show the PLS-R models of the MIR and NIR spectroscopic data, respectively. The PCR analysis of NIR data is depicted in Appendix A. As can be seen in Appendix A, low concentration sample sets show maximum variance and show distinct clustering along PC1 (52.56%) in the PC scores plot. However, more defined clustering can be observed for the 0.5 mg/mL sample set and lower concentrations such as 0.1, 0.05, and 0.01 mg/mL are leading towards the control group in the scores plot. Hence, the possible detection limit is defined as 0.5 mg/mL.

Using data fusion, the regression models were tested to determine whether the combinatorial approach would lead to lower RMSEP values and LoDs than the individual modalities. Table 1 shows the summary of the PLS-R and PCR statistics that were applied in the study. A glycine concentration of 0.01 mg/mL spiked into the serum sample was prepared as a blind sample set and used to optimize the model’s behavior. Figure 4 demonstrates the regression trend using the PLS-R method, including the blind sample set over the 12,500–1350 nm region. The regression curve was able to produce an RMSEP value of 0.318 mg/mL with a perfect predicted R^2^ value of 1.00. The respective regression coefficient shows the bands responsible for the high correlation. Intense regression bands are attributed to CH_2_ rocking (906 cm^–1^), C-N stretching (1034 cm^–1^), N-H_3_ rocking (1111 cm^–1^), C-H_2_ wagging (1337 cm^–1^), or O-C=O symmetric stretching (1414 cm^–1^). We have further investigated the sensitivity of the data fusion model by excluding the blind sample set and checking whether the exhibited tendency is due to the presence of the lowest concentration set (i.e., 0.01 mg/mL) in the data. It is evident from Table 1 that the observed behavior is irrespective of the 0.01 mg/mL data set. The combined spectral data (NIR-ATR) with the blind sample set excluded provided an RMSEP value of 0.303 mg/mL with a predicted R^2^ value of 0.999 and a small difference of 0.015 mg/mL compared to the model that included the lowest concentration (i.e., 0.01 mg/mL) data set. Hence, the models have incorporated the lowest concentration in the series. Similarly, the PCR model over the 12,500–1350 nm region provides an RMSEP value of 0.825 mg/mL with a predicted R^2^ value of 0.997 for 10 PCs (Figure 5). The regression vector shows similar characteristics as depicted in Figure 5e. Interestingly, a detection limit of 0.01 mg/mL was achieved with PC analysis, which is defined as the lowest analyte concentration to reliably distinguish glycine from the limit of blank (LoB). The LoB is the highest analyte concentration expected to be found when replicates of a blank sample containing no analyte are tested [46]. It can be calculated as follows:LoB = mean_blank_ + 1.645 (SD_blank_)(1)
LoD = LoB + 1.645 (SD_low concentration sample_) (2)

SD stands for standard deviation. SD_blank_ refers to the standard deviations for repeated measurements of a blank sample. SD_low concentration sample_ is the standard deviation measurement of a sample known to contain a low concentration of analyte. Mean_blank_, SD_blank_, and SD_low concentration_ values were calculated separately from the measured and predicted values from the regression statistics of ATR, NIR, and combined ATR-NIR spectroscopy. The values were used to analyze LoB and LoD and are shown in Appendix A. By applying the above equations, the LoD was found to be 0.26 mg/mL with ATR spectroscopy and 0.22 mg/mL with NIR spectroscopy. The combined analysis was able to demonstrate an improved detection limit of 0.17 mg/mL. Moreover, the detection limit with ATR spectroscopy was also improved from a previously reported study considering whole serum analysis (0.383 ± 0.007 mg/mL) [32].

## 4. Discussion

Here we show proof of concept for using a miniaturized NIR spectroscopic stand-alone method and a combined mid-IR/NIR data fusion approach to detect a low-molecular-weight compound in serum. The NIR spectroscopic approach was able to provide a detection limit of 0.228 mg/mL, whilst the combined approach was able to show 0.17 mg/mL. The given values are higher than the physiological level of glycine; however, the study shows the potential of NIR spectroscopy and combinatorial analysis of LMW serum biomolecules. ATR spectroscopy has been explored in glycine detection using wet and dry serum samples with the aid of ultra-centrifugal filtration by depleting high-molecular-weight serum components [32]. The regression models on wet serum analysis produced a prediction of a 0.45 ± 0.16 mg/mL glycine sample [32]. Depletion of high-molecular-weight serum components resulted in quantification of glycine levels higher than 0.01 mg/mL, 50 times lower than that of the whole serum. The depletion of the abundant high-molecular-weight serum proteins improved the sensitivity for the detection of glycine and the regression model resulted in a predicted value of 0.011 mg/mL ± 0.006 mg/mL, for the 0.01 mg/mL sample [32]. However, the depletion of high-molecular-weight proteins may result in the specific removal of low abundance cytokines, peptide hormones, and lipoproteins of interest. For instance, albumin is a transport protein that binds a large variety of compounds including hormones, lipoproteins, and amino acids, and depletion of albumin may result in the specific removal of molecules of interest. In comparison to previous ATR spectroscopic studies, our approach has improved the detection limit for glycine in whole serum. However, detecting glycine at its physiological range has not been achieved to date in any reported study. The improved glycine detection limit achieved using the combinatorial spectral approach and the simplicity of the technique provide an advantage over previous studies where a filtration step is required. Future studies will analyze model systems with multispectral modeling involving IR and Raman methods to improve the detection limit of low-molecular-weight proteins in human serum.

## 5. Conclusions

For the first time, multimodal IR spectral combining NIR and ATR-FTIR has been applied for the detection limit of a low-molecular-weight compound in human serum. The multimodal approach incorporating ATR and NIR spectroscopic approaches improves the detection limit of glycine in serum. The LoD was found to be 0.26 mg/mL with ATR spectroscopy and 0.22 mg/mL with NIR spectroscopy. Supervised extended wavelength PLS-R resulted in an RMSEP value of 0.303 mg/mL and an R^2^ value of 0.999 over a concentration range of 0–50 mg/mL of glycine spiked in whole serum. The LoD was improved to 0.17 mg/mL from 0.26 mg/mL. The main limitation of the study would be to be on par with current detection technologies such as antibody assays and liquid chromatography-tandem mass spectroscopy. They are able to detect and quantify the amino acid/protein concentration within the physiological range. Moreover, we employed glycine as a model molecule/system to analyze the detection levels of LMW proteins and amino acids. Hence the results need to be verified with LMW proteins and other amino acids. In the future, we will exploit the proposed combinatorial spectral approach with Raman spectroscopy to assess the LoD of LMW proteins and amino acids. In summary, this proof-of-concept study shows that the detection limit of diagnostically dominant LMWF compounds can be improved using a combined near-IR and mid-infrared spectroscopic approach.

## Figures and Tables

**Figure 1 sensors-22-04528-f001:**
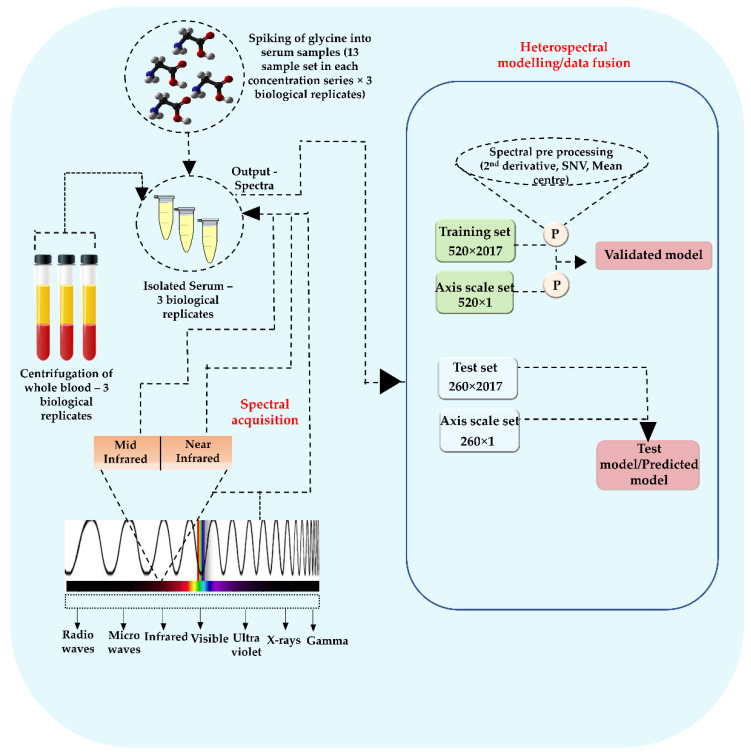
Conceptual representation of experiment and associated chemometrics.

**Figure 2 sensors-22-04528-f002:**
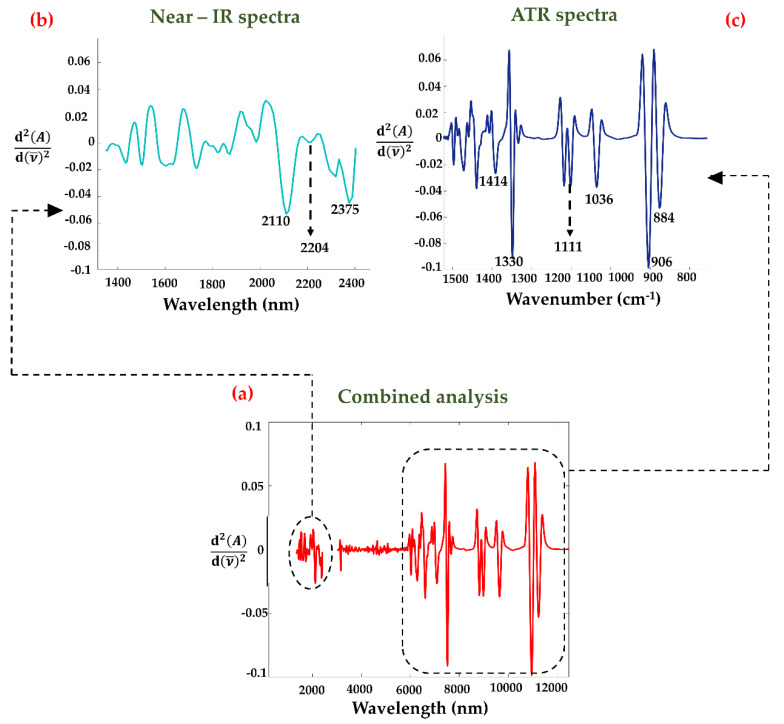
(**a**) Second derivatized spectra of fused ATR–NIR datasets of dried glycine deposits in 12,500–1350 nm region. Magnified view of (**b**) NIR region (2500–1350 nm) and (**c**) mid-IR region (1600–800 cm^−1^). Important bands in the spectra are labeled.

**Figure 3 sensors-22-04528-f003:**
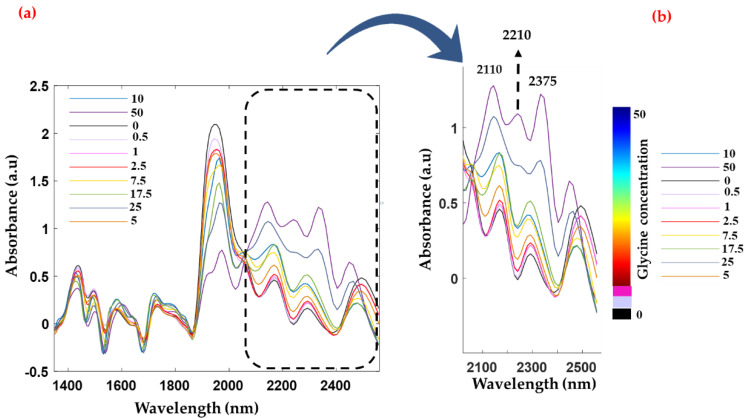
(**a**) Baseline corrected raw spectra of glycine-spiked human serum samples. The concentration of glycine ranges from 0–50 mg/mL. (**b**) Magnified view of 2500–2000 nm region where major contribution from glycine is observed.

**Figure 4 sensors-22-04528-f004:**
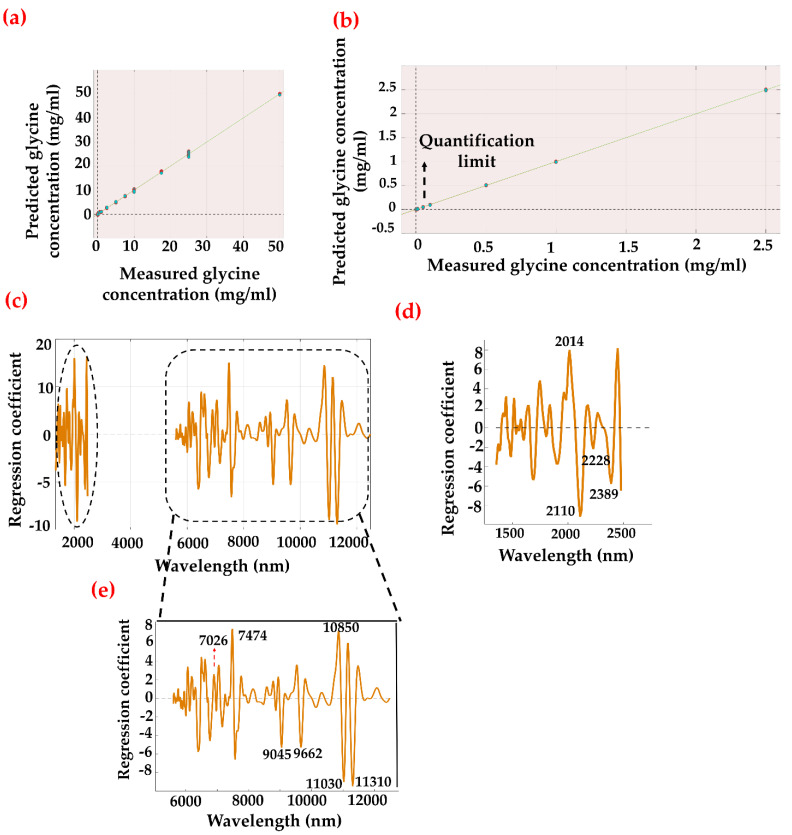
PLS-R predicted model of glycine-spiked serum samples. (**a**) Regression plot of entire range between 0 (control) and 50 mg/mL region. (**b**) Extrapolated and remodeled 0–2.5 mg/mL region. (**c**) Corresponding regression coefficient/vector over hetero-spectral region covering NIR (2500–1350 nm) and MIR (12,500–5555 nm). (**d**) Magnified view of regression vector over 2500–1350 nm and (**e**) 12,500–5555 nm region.

**Figure 5 sensors-22-04528-f005:**
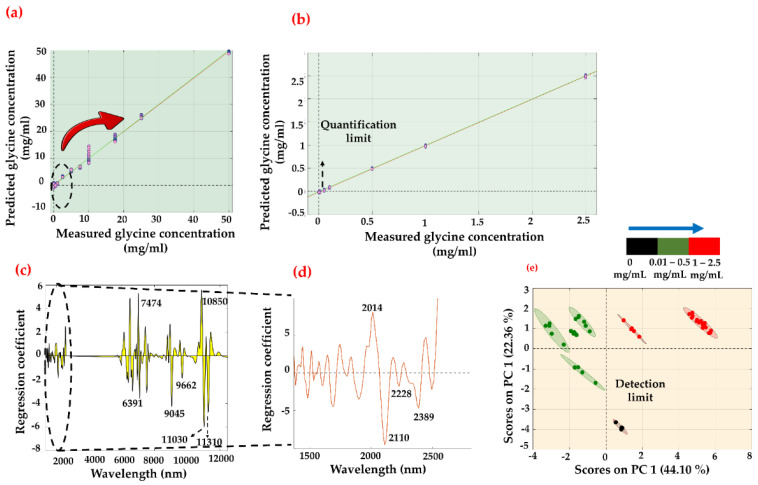
PCR predicted model of glycine-spiked serum samples. (**a**) Regression plot of entire range between 0 (control) and 50 mg/mL region. (**b**) Extrapolated and remodeled 0–2.5 mg/mL region. (**c**) Corresponding regression coefficient/vector over hetero-spectral region covering NIR (2500–1350 nm) and MIR (1800–800 cm^−1^). (**d**) Magnified view of regression vector over 2500–1350 nm and (**e**) scores plot between PC1 and PC2.

**Table 1 sensors-22-04528-t001:** Summary of regression statistics.

Spectroscopic Technique	Regression Method	Region (nm)	No. of LVs/PCs	RMSEC	RMSECV	RMSEP	R^2^
NIR	Partial-least squares (PLS)	2550–1350	8	0.324	0.34	0.3918	0.999
NIR	Principal component regression (PCR)	2550–1350	8	0.76	0.83	0.866	0.997
ATR	Partial-least squares (PLS)	12,500–5550	10	0.72	0.77	0.7238	0.997
Combined spectral data (NIR-ATR)	Partial-least squares (PLS) (Including blind sample set)	2550–1350 12,500–5550	10	0.328	0.356	0.318	1.000
Combined spectral data (NIR-ATR)	Partial-least squares (PLS) (Excluding blind sample set)	2550–1350 12,500–5550	10	0.307	0.344	0.303	0.997
Combined spectral data (NIR-ATR)	Principle component regression (PCR) (Including blind sample set)	2550–2000 12,500–5550	10	0.872	0.9	0.825	0.997
Combined spectral data (NIR-ATR)	Principle component regression (PLS) (Excluding blind sample set)	2550–2000 12,500–5550	10	0.747	0.861	0.688	0.997

## Data Availability

Not applicable.

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
