# Peer review of "A Combined Near-Infrared and Mid-Infrared Spectroscopic Approach for the Detection and Quantification of Glycine in Human Serum"

_sensors, 2022, doi:10.3390/s22124528_

Round 1

Reviewer 1 Report

In this article, the authors claim that the analysis of combined NIR and MIR spectroscopic data helps to improve the limit of detection. They have used glycine (as a sample analyte) in serum for their proof of concept study. Overall, it’s a nice study but the presentation of the results/graphs must be improved (Figures need better labeling, legend, and caption). The author should explain more clearly how they obtained the RMSEP values from the presented data. Similarly, the authors should provide more explanation of the parameters in Eq.1 and 2, as well as the values used for calculating LoD. 

Author Response

See attached file for Reviewer 1

Reviewer 2 Report

The authors presented a new proof of concept, they used a combined near-infrared and mid-infrared spectroscopic approach to detect low-molecular-weight glycine. It is a very good idea. However, there are some defects in the experiment design.

1. The number of serum samples is too less, there were only 3 serum samples used to prepare 39 samples for this study. The human serum samples are quite different among people, especially for ill people, it will have a significant influence on the IR and NIR spectrum, and then affect the calibration performance.

2. The RMSEP used in this study was actually for the calibration set. Usually, RMSEC and RMSECV should be used. therefore,  it (RMSEP used in this study) cannot demonstrate the method's performance. The performance of an analysis method should be validated with a set of new samples. However, I have not seen it in this work,

Author Response

Please see attached file for Reviewer 2's comments

Author Response

Please see attached file for the response to reviewer 3's comments

Round 2

Reviewer 1 Report

correct label Figure S4

Reviewer 2 Report

All issues have been resolved.